# Dual sensory impairments in companion dogs: Prevalence and relationship to cognitive impairment

Ryan G. Hopper[1☯], Rachel B. Bromberg[1☯], Michele M. Salzman[1], Kyle D. Peterson[2], Callie Rogers[1], Starr Cameron[3], Freya M. Mowat[1,2]*

1 Department of Surgical Sciences, School of Veterinary Medicine, University of Wisconsin-Madison, Madison, Wisconsin, United States of America, 2 Department of Ophthalmology and Visual Sciences, School of Medicine and Public Health, University of Wisconsin-Madison, Madison, Wisconsin, United States of America, 3 Department of Medical Sciences, School of Veterinary Medicine, University of Wisconsin-Madison, Madison, Wisconsin, United States of America

☯ These authors contributed equally to this work.
* mowat@wisc.edu

## Abstract

### Purpose

Many older dogs (*Canis lupus familiaris*) develop cognitive impairment. Dog owners often describe impairments in multiple sensory functions, yet the relationships between sensory and cognitive function in older dogs is not well understood.

### Methods

We performed assessments of dog vision and hearing, both clinically (n = 91, electroretinography and brainstem auditory evoked potential) and via validated questionnaire (n = 238). We determined prevalence of sole and dual hearing/vision impairments in younger (<8 years) and older (≥8 years) dogs. Impairment cutoffs were determined using data from young dogs. We assessed the relationships between questionnaire-assessed vision and/or hearing impairments and cognitive impairment using logistic regression.

### Results

Younger and older dog groups had similar distributions of sex and purebred/mixed breed status. Sex had no relationship to prevalence of sensory impairments. Older dogs had higher prevalence of hearing, vision, and dual sensory impairments, assessed both clinically and by questionnaire (P<0.001), and cognitive impairment assessed by questionnaire (P<0.001). Dogs had higher prevalence of reported cognitive impairment when owners reported dual vision and hearing impairments (79–94%, versus 25–27% in dogs with no sensory impairments), which was most consistent in dogs aged ≥8 years. In these older dogs, dual vision/hearing impairments were associated with a significantly increased risk of cognitive impairment (1.8–2.0 odds ratio).

**Data Availability Statement:** All relevant data are within the manuscript and Supporting information files.

**Funding:** This work was funded by the National Institutes of Health (K08EY028628, R01AG082907, P30EY016665), the Morris Animal Foundation Mark L. Morris Jr. Investigator Award (D23CA-510), donor funds to the UW-Madison Department of Ophthalmology and Visual Sciences, an unrestricted grant from Research to Prevent Blindness, Inc. to the UW-Madison Department of Ophthalmology and Visual Sciences, and the UW-Madison Women in Science and Engineering and Leadership Institute Vilas Life Cycle Professorship. Funders had no role in study design, data collection and analysis, decision to publish, or preparation of the manuscript.

**Competing interests:** The authors have declared that no competing interests exist.

## Conclusion

Dogs aged ≥8 years are at higher risk for dual hearing/vision impairments and associated cognitive impairments. The causal relationship between these impairments is not defined, but clinical consideration of these multimorbidity risks should be made in older dogs.

## Introduction

Cognitive impairment in companion dogs is a progressive neurodegenerative disorder characterized by dementia-like behavioral changes [1]. The prevalence of cognitive impairment is age-associated as shown in multiple studies [2–6]; reported prevalence is as high as 68% in 15–16-year-old dogs [3]. The age of 8 years appears to be an important age at which risk for cognitive impairment increases in dogs [1]: in one study of dogs over 8 years examined over a period of 2 years, 33% of dogs with normal cognitive function developed mild cognitive impairment, and 22% of dogs with mild cognitive impairment progressed to severe cognitive impairment [7]. The disease in dogs has strong parallels with human Alzheimer's disease and related dementias (AD/ADRD). Notable dog/human similarities include clinical manifestation of signs including reduced social interaction and spatial awareness, confusion, irritability, and disrupted sleep patterns [1, 7–9]. There are also notable similarities between dog and human pathology associated with dementia including perivascular deposition of Amyloid-beta [10–12], neuroinflammation [11], and hyperphosphorylated tau accumulation [11]. With so many pathophysiologic similarities to humans and a large potential study population, companion dogs may provide a relevant, time-efficient sentinel species in which to study risk factors for AD/ADRD in humans.

Risk factors for canine cognitive impairment are emerging, and while nutrition and body condition [2, 13, 14] have been implicated as possible risk factors, multiple studies have also drawn a connection between sensory impairment and cognitive impairment in dogs [4, 5, 15]. In dogs, age increases the risk of both vision impairment [16, 17] and hearing impairment [18–20], although general population prevalence of these sensory impairments has not been reported. Despite extensive epidemiological study, the causative relationship between human sensory impairments and cognitive impairment remains unclear. Patients with dementia are at increased risk of developing impairments in vision [21] and hearing [22]. Conversely, there is a higher risk of dementia in patients with sensory morbidities including vision impairment [23–28], hearing impairment [28–31], and olfactory impairment [32, 33]. In fact, dual sensory impairment of both hearing and vision increases the risk of cognitive impairment by 8-fold [28], more than what is expected from the sum of each individual factor. This finding indicates that sensory impairments are multiplicative rather than additive in terms of risk for cognitive impairment. To-date, analysis of cognitive and sensory relationships in companion dogs are limited to a single sense [4, 15, 34]. Concurrent study of sensory and cognitive outcome measures is advocated for in human dementia research [35].

We conducted a study to determine the prevalence of sensory impairments of hearing and vision in dogs, either as sole impairments or in combination, and to relate these results to prevalence of cognitive impairment. We performed studies using both clinical and questionnaire-based outcome measures.

## Materials and methods

### Institutional approval

Use of the questionnaire-based proxy assessment was evaluated and exempted by the Institutional Review Board of the University of Wisconsin-Madison and the UW-Madison Institutional Animal Care and Use Committee. Validation in the subset of dogs in which electroretinography and brainstem auditory evoked potentials were performed was approved by the University of Wisconsin-Madison Institutional Animal Care and Use Committee (approval number V006521). All dog owners signed a written informed consent form prior to enrollment of their dog.

### Study participants

Questionnaires were distributed physically (mail or direct) or digitally to potential dog owners who were invited to participate in a study of dog aging based in Wisconsin. Requests for questionnaire completion were sent to n = 1,935 potential participants, of which n = 46 were known dog owners and n = 1,889 were unknown as to their dog owning status. Participants were solicited via direct request to known prior participants in research studies in the university, and via fliers sent to local veterinary practices. Eligibility to participate was open to people that cohabited with a dog. If participants had multiple dogs, they were instructed to respond based on the oldest, healthiest dog in the household. A subset of dog owners was invited to bring their dog for a clinical evaluation at the University of Wisconsin-Madison School of Veterinary Medicine. Inclusion criteria for the clinical evaluation were age greater than 1 year and health/temperament suited to study procedures without administration of anxiolytic or sedative medications. These medications are shown to impact neurologic responses [36] and retinal function [17, 37] in dogs.

### Questionnaire-based assessment

All participating owners were asked to complete a questionnaire within 1 month of a clinical visit with their dog, if one was scheduled. Information collected included demographics (date of birth or approximate age if date unknown, breed if purebred, sex, neutering status, body weight category), and standardized questionnaire responses regarding dog hearing (the 8-question dog hearing function questionnaire; HFQ [38]), vision (the 17-question dog variable lighting questionnaire; dogVLQ [16]), and cognitive impairment (the 17-question CAnine DEmentia Scale; CADES [1]). These questionnaires have been extensively validated against objective measures in prior studies. In the original report, the HFQ was validated in dogs with otitis externa or otitis media, using brainstem auditory evoked potential threshold testing (similar to our methodology described below) [38]. The recently reported dogVLQ was extensively validated in healthy dogs against the gold-standard, objective measure of retinal function, electroretinography [16]. The CADES is extensively validated against other canine cognitive questionnaires [1], clinical cognitive tests [15, 39], and neurodegeneration biomarkers [39–42]. Scores for each function questionnaire were manually reviewed and responses excluded if incomplete data were obtained. The HFQ was converted to a percentage (100% relating to optimal hearing), to compare with the dogVLQ which is designed on a 0–100 scale. Cutoffs for impairment for the sensory function questionnaires were calculated as described below. Cutoffs for cognitive impairment were utilized as published [1].

### Clinical assessment

A subset of dogs that had owner questionnaire data obtained (n = 91) participated in a clinical assessment, which included a full ophthalmic examination by a board-certified veterinary

ophthalmologist and a neurologic exam by a board-certified veterinary neurologist. Dogs also underwent a routine physical examination, including an otoscopic examination. Based on the results of the ophthalmic and otoscopic examination, one eye and one ear (with the fewest abnormalities) were selected for electrophysiologic testing (electroretinogram and brainstem auditory evoked potential testing, respectively). If eyes or ears had similar numbers of abnormalities (or no abnormality), the ear or eye to be tested was selected at random (Randomizer. org). Dogs were not administered any sedation or anxiolytics, and tests were omitted if dogs did not tolerate testing with gentle manual restraint.

Brainstem auditory evoked potential (BAEP) testing was performed unilaterally (Cadwell Sierra II Wave console and Sierra II Wave 4 Channel Amplifier, Sierra Wave Software 11.0.116, Cadwell Laboratories Inc. Kennewick, WA). As previously described [38, 43], platinum subdermal needle electrodes (Natus Neurology, Middleton, WI), were placed halfway between the occiput and top of shoulder blades (ground), at the midline vertex (active/positive), and adjacent to the tragus ipsilateral to the tested ear (reference). Foam tip tubal inserts (size 10mm, 13mm, or 18mm depending on the size of the dog's ear canal) were placed binaurally (Etymotic Research, Elk Grove Village, IL). Impedance was maintained at $<5 \, k\Omega$. Stimuli were broadband clicks presented at 11.33 Hz. Responses were filtered 30–3,000 Hz and averaged. A broadband masking noise of 30 dB nHL below the tested ear stimulus intensity was presented in the contralateral ear. The test protocol included the following stimuli: 90dB normal hearing level (nHL; 1000 clicks averaged), 70 dB nHL and 50dB nHL (500 clicks averaged). After this series of testing, dogs underwent threshold testing which included reduction in stimulus intensity in 20dB nHL increments until wave V was no longer visible, at which point stimulus intensity was then increased by 5dB increments until wave V was again observed (minimum of 250 clicks averaged), as previously described [38, 43]. The BAEP threshold was recorded as the lowest stimulus intensity that elicited a detectable wave V, resolution was within +/-5dB nHL. To estimate the relationship between our presented stimulus intensity (in nHL units) to standard stimulus intensity (sound pressure level; SPL), we calculated a conversion. Click stimuli nHL was calibrated to peak-to-peak equivalent SPLs based on published recommendations [44]. The process involved comparing peak-to-peak voltages of the clicks transmitted via insert earphones to those of a 1000 Hz pure tone. Subsequently, SPL were derived by cross-referencing the output in SPL for the pure tone using a sound level meter (model 824, Larson and Davis, Depew, NY).

Electroretinography (ERG) was performed as previously described [17], unilaterally following mydriasis (Tropicamide 1%, Akron, Lake Forest, IL, administered topically a minimum of 40 minutes prior to testing) using a handheld device (RetEval, LKC Technologies, Gaithersburg, MD). Briefly, a gold foil corneal contact electrode (ERG-Jet, Fabrinal SA LA, Chaux-De-Fonds, CH) was placed with topical ophthalmic local analgesia (Proparacaine, Alcon Laboratories, Fort Worth TX) and coupling gel (Genteal Gel for Severe Dry Eye, Alcon Laboratories, Fort Worth TX). Platinum needle skin electrodes (identical to those used for BAEP) were placed 3cm lateral to the lateral canthus (reference) and at the occiput (ground). Collection of bright flash responses began after a minimum of 10 minutes of exposure to overhead room lighting (range from 107–120 cd/m$^2$) and with a 30cd/m$^2$ background light. This was followed by 20 minutes of dark adaptation and subsequent collection of dark-adapted responses under dim red-light illumination.

## Statistical analysis

To account for potential variance in lifespan between smaller ($<50$kg) and large ($>50$kg) dogs [45], all purebred dogs and all dogs that had a clinical examination performed were assigned a

life stage, using standardized criteria [46]. Mixed breed dogs were assigned an estimated life-stage based on body size, body weight, and conformation. Dogs were classified as juvenile (< 18 months, lifestage 1), young adult (≥18 months but less than 50% anticipated life expectancy, lifestage 2), mature adult (50–75% anticipated life expectancy, lifestage 3), senior (last 25% of anticipated life expectancy, lifestage 4), and geriatric (exceeding anticipated life expectancy, lifestage 5). No dogs of lifestage 1 underwent a clinical examination, and very few were described in the questionnaire dataset (n = 4), therefore statistical analysis involving lifestage did not include dogs from lifestage 1.

Impairment cutoffs for clinical electrophysiologic tests (BAEP threshold, ERG b-wave amplitude) and sensory function questionnaires (HFQ, dogVLQ) have not been established. We established cutoffs using our most numerous group of young dogs (dogs at lifestage 2). Questionnaire responses and clinical outcome measures were used to calculate the mean value for dogs at lifestage 2, and cutoffs were defined as mean +/- 1.96 standard deviations as previously described [47]. Details of cutoffs are provided in Table 1. We utilized the amplitudes of the b-wave, as this is the ERG outcome most associated with age in dogs [17]. Out of the ERG flash intensities, the 10cds/m$^2$ light-adapted (representing a cone photoreceptor isolated response) and 0.1cds/m$^2$ dark-adapted (representing a mixed rod-cone photoreceptor response) outcomes were used in analysis.

Using these impairment cutoffs, we estimated the sensitivity and specificity of the function questionnaires to detect impairment using the objective clinical tests using standard methods [48]. Briefly, we calculated true and false positive and negative rates of impairment based on the function questionnaires compared with the gold standard clinical tests; sensitivity was calculated as the number of true positives divided by the sum of the true positives and the false negatives; specificity was calculated as the number of true negatives divided by the sum of the true negatives and the false positives. We also calculated the Youden index of the questionnaires using standard methods (sensitivity plus specificity, minus 100) [49]. Descriptive statistics were generated by cross-tabulating sensory impairment with sex, age, and weight, and were analyzed for differences using Fisher exact tests. Prevalence of cognitive impairment across sex, age, and weight were also analyzed using Fisher exact tests. Associations of cognitive impairment for dogs aged 8 years or older were estimated with odds ratio and 95% confidence intervals from logistic regression models fit using hearing and/or vision impairments as covariates. Separate models were fit for bright versus dark vision, respectively. Statistical testing was performed in GraphPad Prism (version 10.1 for Mac), and R (version 4.2.3). Significance was determined a priori with alpha set to 0.05 for all tests.

## Results

### Demographics

A total of 238 dog owners submitted a completed questionnaire, 91 of those dogs underwent a clinical visit. The demographics of the participants (questionnaire and clinical) is detailed in Table 1.

### Clinical examination findings

Neurologic examination: 32 of the 91 dogs had at least 1 neurologic abnormality identified on neurologic examination. No neurologic abnormality was interpreted as severe. Neurologic abnormalities included cranial nerve deficits (n = 4; anisocoria attributed to iris atrophy), postural reaction deficits (n = 25), ataxia (n = 18), paresis (n = 14), abnormal withdrawal reflex (n = 3), abnormal patellar reflexes (n = 8), pain on spinal palpation (n = 9), pain on tail range of motion (n = 2), and pain on cervical range of motion (n = 2). Neuroanatomic localization

**Table 1. Median, interquartile ranges for demographics of dogs for clinical evaluation and for questionnaire analyses.**

| | Clinical evaluation subset (n = 91) | All dogs with questionnaire data (n = 238) |
|---|---|---|
| Age (months) | 89 (61–123) | 95 (60–133) |
| % aged < 8 years | 53.9% | 50.8% |
| Lifestage, % (n) | | |
| 1 | NA | 2.6% (4) |
| 2 | 26.4% (24) | 23.8% (36) |
| 3 | 27.4% (25) | 23.8% (36) |
| 4 | 24.2% (22) | 28.4% (43) |
| 5 | 22.0% (20) | 21.2% (32) |
| Breed (% purebred) | 53.9% | 48.3% |
| Sex (% female) | 50.5% | 51.7% |
| % spayed/neutered | 94.5% | 93.2% |
| Body weight (lb) | 38.3 (21.9–57.6) | NA |
| % <50lb | 41.8% | 52.5% |
| Cephalic index | 47.8 (42.9–51.7) | NA |
| BAEP parameters (n = 82) | | NA |
| Threshold (dB nHL) | 10 (5–20) | |
| Cutoff for impairment (dB nHL), n = 19 | 22.5 | |
| Number of stimuli presented, mean (SD) | | |
| 90dB | 995 (64) | |
| 70dB | 539 (177) | |
| 50dB | 587 (242) | |
| 90dB amplitude (μV) | | |
| wave I | 2.13 (1.58–3.18) | |
| wave V | 1.65 (1.20–2.43) | |
| 90dB latency (ms) | | |
| wave I | 0.94 (0.90–0.97) | |
| wave V | 3.39 (3.29–3.48) | |
| 70dB amplitude (μV) | | |
| wave I | 1.03 (0.57–1.62) | |
| wave V | 1.62 (1.15–2.18) | |
| 70dB latency (ms) | | |
| wave I | 1.05 (1.00–1.11) | |
| wave V | 3.50 (3.41–3.61) | |
| 50dB amplitude (μV) | | |
| wave V | 1.31 (0.99–2.04) | |
| 50dB latency (ms) | | |
| wave V | 3.67 (3.52–3.83) | |
| ERG parameters (n = 87) | | NA |
| LA10 b-wave amplitude (μV) | 74.1 (55.7–85.2) | |
| Cutoff for impairment (μV), n = 22 | 48.8 | |
| DA0.1 b-wave amplitude (μV) | 214.8 (171.0–263.3) | |
| Cutoff for impairment (μV), n = 21 | 132.1 | |
| HFQ score (%) | 87.5 (87.5–100) | 88 (88–100) |
| Cutoff for impairment (%), n = 36 | | 76.1 |
| dogVLQ score | | |

(*Continued*)

**Table 1.** (Continued)

| | Clinical evaluation subset (n = 91) | All dogs with questionnaire data (n = 238) |
|---|---|---|
| bright lighting | 100 (96.9–100) | 100 (97.20–100) |
| Cutoff for impairment, n = 36 | | 94.2 |
| near darkness | 100 (89.4–100) | 100 (87.5–100) |
| Cutoff for impairment, n = 35 | | 77 |
| CADES | | |
| Score | 3 (0–9) | 4 (0–10) |
| Category | 0 (0–1) | 0 (0–1) |
| % any impairment | 27.5% | 32.6% |

findings included C1-C5 myelopathy (n = 3), T3-L3 myelopathy (n = 13), S4-caudal myelopathy (n = 3), and T3-caudal myelopathy (n = 2). One dog was neurolocalized to either a C1-C5 myelopathy or intracranial neurologic disease based on the presence of unilateral thoracic and pelvic postural placing deficits (not included in the above neurolocalization counts). Seventy dogs were considered to have a normal neurologic examination (i.e. minimal to no deficits), 11 of which had one or more abnormalities on neurologic exam, such as postural placing deficits, pain on palpation, and mild paraparesis.

Ophthalmic examination: All dogs were deemed as visual by the presence of a menace response. No dog had evidence of uveitis. Median intraocular pressure in the eye that underwent ERG testing was 19mmHg (interquartile range 16-21mmHg). Clinical findings in the ERG tested eye included focal epithelial pigmentation (n = 1), stippling of the epithelial surface (n = 3), focal stromal opacity (n = 2), mild corneal edema (n = 2), and focal endothelial pigmentation (n = 2). Clinical findings in the lens included nuclear sclerosis (n = 76), nuclear fibrillar changes (n = 16), focal capsular opacity (n = 4), incipient cortical cataract (n = 19), mesenchymal pigment remnants on anterior capsule (n = 4), early immature cataract (n = 1), and early mature cataract (n = 1). Clinical findings of the vitreous included asteroid hyalosis (n = 1) and vitreal degeneration/syneresis (n = 5). Clinical findings of the retina included mottled tapetum (n = 5), mild retinal vascular attenuation (n = 2), absent tapetum (n = 2) focal areas of tapetal hyper-reflectivity (n = 1), pigment clumping in nontapetal fundus (n = 1), areas of pigment in tapetal fundus (n = 1), multifocal areas of altered tapetal reflectivity (n = 1), peripapillary conus (n = 1), choroidal hypoplasia (n = 1), discolored tapetum (n = 1), and possible focal retinal hemorrhage (n = 1). The blood pressure for the dog with possible hemorrhage was elevated (mean systolic blood pressure 202mmHg). The retina of one eye could not be evaluated in detail due to the presence of cataract and two dogs had a reduced amount of RPE/choroidal pigmentation (subalbinotic). A total of 41 left and 50 right eyes were selected for electrodiagnostic testing. No dogs were excluded from performing ERG because of clinical examination findings.

Otoscopic examination: Not all dogs had all parts of the otoscopic examination completed due to compliance. Clinical examination findings in ears selected for BAEP included pruritis (n = 14), pinnal dermatitis (n = 15), palpable aural thickening (n = 8), material in the external ear canal (n = 21), erythema (n = 12), and varying degrees of partial external canal occlusion (n = 11). Of the ears that underwent otoscopic examination, the tympanum was fully visualized in 76 cases and partially obscured in 2 cases. The tympanum appearance was normal in 75 cases, mildly abnormal in 2 cases, and not described in 1 case. A total of 62 left ears and 29

right ears were selected for electrodiagnostic testing. No dogs were excluded from performing BAEP because of clinical examination findings.

## Hearing function evaluation

Clinical BAEP testing: Associations between age and BAEP amplitudes and latencies are shown in S1 and S2 Figs, summary data are presented in Table 1. There was a significant negative association between age and amplitude of wave I (90dB nHL; Spearman r -0.47, $p < 0.0001$) and wave V (70dB nHL Spearman r -0.25, $p = 0.03$, and 50dB nHL Spearman r -0.38, $p = 0.0006$). There was a significant positive association between age and latency of wave V (90dB nHL Spearman r 0.25, $p = 0.02$, 70dB nHL Spearman r 0.24, $p = 0.03$). Higher BAEP threshold was associated with dog age (Spearman r 0.53, $p < 0.0001$; Fig 1A) and assigned dog life stage (geriatric; Fig 1B). Cutoff for impairment was calculated as > 22.5dB nHL (Table 1). S3 Fig shows the estimated SPL level based on the original nHL level. Given the linear change in intensity levels, estimates of thresholds and cutoffs can be estimated based on these data using the equation $Y = 1.017*X + 24.14$. Based on our conversion data the cutoff for impairment in SPL is estimated to be 47dB SPL. Of the dogs that exceeded this threshold (n = 11), 9/11 (82%) were aged 8 years or older. The prevalence of any hearing deficit (sole or combined) in dogs aged less than 8 years (4.8%) was lower than that in dogs aged 8 years or older (24.3%).

Questionnaire-assessed hearing function: Summary data are presented in Table 1. There was a significant negative association between age and hearing function questionnaire score (Spearman r -0.29, $p < 0.0001$, Fig 1C). Similarly, dog assigned lifestage was significantly associated with worse hearing function score (geriatric; Fig 1D). Cutoff for impairment was calculated as a score < 76.1 (which equated to at least 2/8 questionnaire responses indicating hearing loss; Table 1). Of the dogs that exceeded this threshold (n = 46), 32/46 (70%) were aged over 8 years. The prevalence of any hearing deficit (sole or combined) in dogs aged less than 8 years (11.6%) was lower than that in dogs aged 8 years or older (27.4%). Because the original report of the HFQ was validated in dogs clinically affected by otitis [38], we performed association analysis between the HFQ and BAEP threshold (Fig 1E) to validate it further in dogs with healthy ears. The outcomes of the hearing function questionnaire (HFQ) were significantly associated with the hearing threshold as estimated clinically using the BAEP threshold (Fig 1E; Spearman r -0.27, p = 0.02). Using the clinical cutoff of 22.5dB nHL (47dB SPL), the HFQ had a sensitivity of 36.4%, specificity of 90.5%, and a Youden index of 0.28. If the cutoff was adjusted to >40dB nHL (64.8dB SPL; consistent with a moderate or worse hearing loss in a previously published study in dogs with otitis [38]), this resulted in improved sensitivity (75%) while retaining high specificity (90.1%), and an improvement in the Youden index to 65%.

## Visual/retinal function evaluation

Electroretinography: Our previous research indicated that ERG b-wave amplitude has the largest association with age in dogs [17], and we selected one dark-adapted and one light-adapted b-wave to calculate cutoffs. Summary data are shown in Table 1. For these stimulus intensities, ERG b-wave amplitudes in both dark-adapted and light-adapted conditions were lower in older dogs (age/lifestage data shown in S4 Fig) and ERG b-wave peak times in both dark-adapted and light-adapted conditions were prolonged in older dogs (S4 Fig). ERG cutoffs for impairment were calculated as < 48.8 μV (light-adapted) and <132.1 μV (dark-adapted; Table 1). Of the dogs that exceeded this threshold (n = 11 light-adapted, 7 dark-adapted), 10/11 (90.9%) light-adapted, and 6/7 (85.7%) dark-adapted were aged 8 years or older. Prevalence

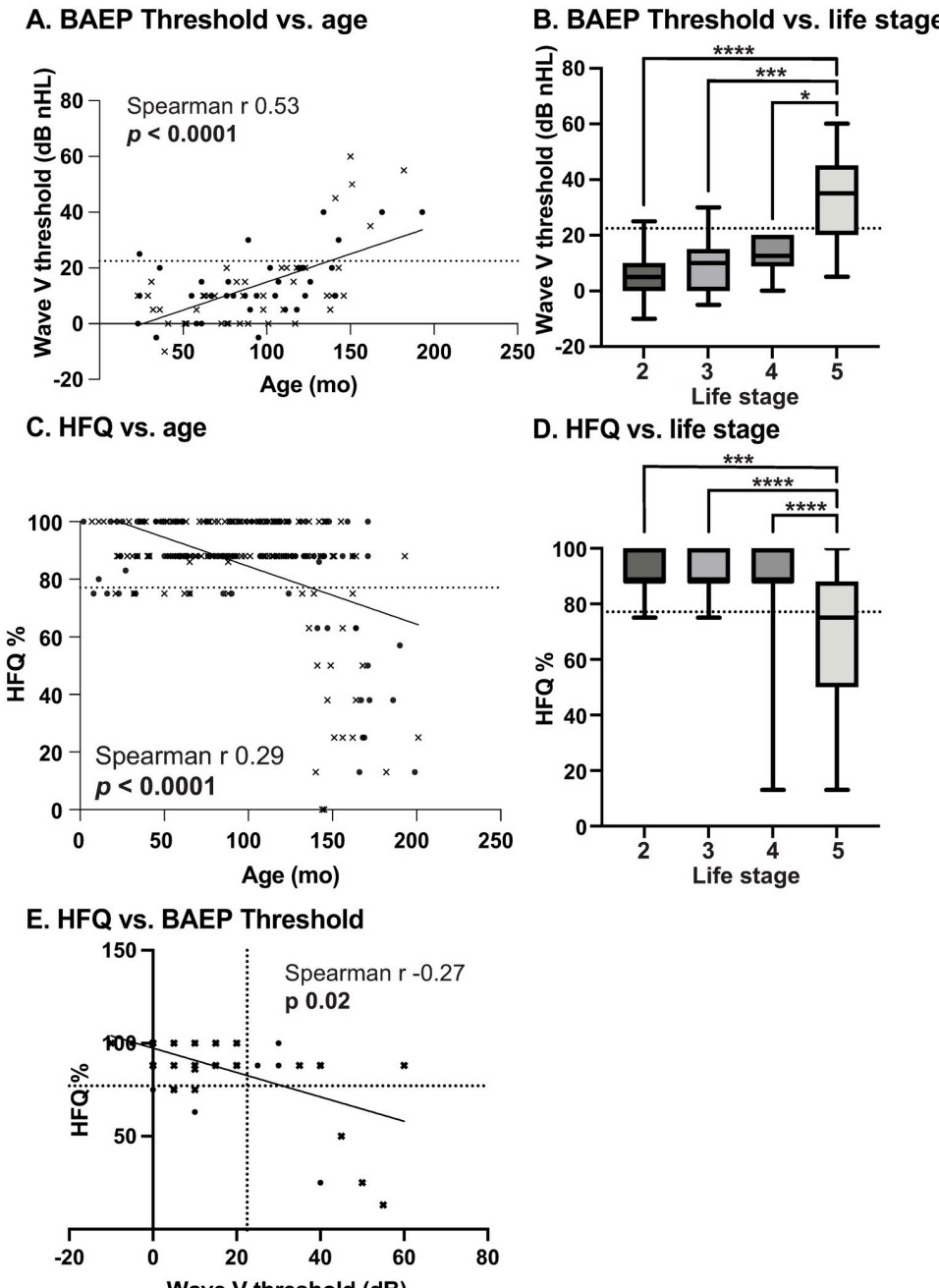

**Fig 1. Age associations and magnitude of age-related hearing loss in dogs evaluated clinically and via owner questionnaire.** Clinically detectable age-related hearing loss was identified by the determining the threshold of detection of wave V in brainstem auditory evoked potential recordings (A). Significant hearing loss was present in dogs at life stage 5 (B). Owner questionnaire detected age-related hearing loss was determined by the hearing function questionnaire (C). Significant hearing loss was present in dogs at life stage 5 (D). Clinical and questionnaire detected hearing function outcomes were associated (E). Spearman rank nonparametric correlation r and p values are shown, regression lines are shown for illustration purposes only. Male dogs are illustrated as crosses, female dogs as circles. Kruskal Wallis test was used for group-based analysis. Dotted line represents impairment cutoff calculated using normative data from dogs at lifestage 2. * indicates p-value <0.05. *** indicates p-value <0.001. **** indicates p-value<0.0001.

of any visual impairment in dogs aged less than 8 years (2.2% light-adapted, 2.3% dark-adapted) was lower than that in dogs aged 8 years or older (23.8% light-adapted, 14.3% dark-adapted). Prevalence of dual hearing and vision impairment in dogs aged less than 8 years (0% light-adapted+BAEP, 0% dark-adapted+BAEP) was lower than that in dogs aged 8 years or older (13.7% light-adapted+BAEP, 7.1% dark-adapted+BAEP).

Visual function questionnaire: Our previous research indicated that visual function questionnaire scores for dog behavior in bright light and near darkness are associated with age in dogs [16] (age/lifestage data shown in S4 Fig). Summary data are shown in Table 1. Visual function questionnaire cutoffs for impairment were calculated as scores < 94.2 (bright lighting) and < 77.0 (near darkness; Table 1). Of the dogs that exceeded this threshold (n = 35 dogVLQ bright, 31 dogVLQ dark), 30/35 (85.7%) for dogVLQ bright, 26/31 (83.9%) dogVLQ bright were aged 8 years or older. Prevalence of any visual deficit (sole or combined) in dogs aged less than 8 years (9.9% dogVLQ bright, 4.2% dogVLQ dark) was lower than that in dogs aged 8 years or older (25.6% dogVLQ bright, 23.8% dogVLQ dark). Prevalence of dual questionnaire-assessed vision and hearing impairments in dogs aged less than 8 years (0.8% dogVLQ bright+HFQ, 0.8% dogVLQ dark+HFQ) was lower than that in dogs aged 8 years or older (16.0% dogVLQ bright+HFQ, 14.3% dogVLQ dark+HFQ).

As previously described [16], outcomes of the visual function questionnaire were associated with the electroretinography b-wave amplitude (data not shown). Using the clinically determined cutoffs for dark- and light-adapted electroretinography, the visual function questionnaire had relatively low sensitivity but high specificity (dogVLQ dark sensitivity 57.1%, specificity 92.3%; dogVLQ bright sensitivity 27.3%, specificity 89.5%). Youden index was higher for the dogVLQ dark (49%) than the dogVLQ bright (17%).

## Prevalence of hearing and visual dysfunctions

Using the determined impairment cutoffs we estimated the prevalence of clinical hearing/ vision impairments (Table 2 light-adapted and Table 3 dark-adapted) and questionnaire-assessed hearing/vision impairments (Table 4 bright light visual behavior and Table 5 visual behavior in near darkness) in relation to demographic covariables (sex male or female, body weight < or ≥ 50lb, and age < or ≥ 8 years). Lifestage and cognitive impairment as defined by

**Table 2. Clinical evaluation of the prevalence of hearing and light-adapted retinal response.** Clinically evaluated characteristics of dogs no sensory impairments, sensory impairments (hearing, light-adapted retinal response) and dual sensory impairments (n = 76).

| | Total (%) | Sensory impairment | | | | |
|---|---|---|---|---|---|---|
| | | No sensory impairment (%) (n = 59) | Sole hearing impairment (%) (n = 7) | Sole light-adapted vision impairment (%) (n = 6) | Dual hearing and vision impairment (%) (n = 4) | P-value |
| Sex | | | | | | |
| Male | 53.9 | 52.5 | 42.9 | 66.7 | 75.0 | 0.75 |
| Female | 46.1 | 47.5 | 57.1 | 33.3 | 25.0 | |
| Body weight | | | | | | |
| <50lb | 34.2 | 39.0 | 14.3 | 16.7 | 25.0 | 0.53 |
| ≥ 50lb | 65.8 | 61.0 | 85.7 | 83.3 | 75.0 | |
| Age | | | | | | |
| <8 years | 51.3 | 61.0 | 28.6 | 16.7 | 0.0 | **0.01** |
| ≥ 8 years | 48.7 | 39.0 | 71.4 | 83.3 | 100.0 | |

**Table 3. Clinical evaluation of the prevalence of hearing and dark-adapted retinal response.** Clinically evaluated characteristics of dogs with no sensory impairments, sensory impairments (hearing, dark-adapted retinal response) and dual sensory impairments (n = 75).

| | Total (%) | Sensory impairment | | | | |
| --- | --- | --- | --- | --- | --- | --- |
| | | No sensory impairment (%) (n = 59) | Sole hearing impairment (%) (n = 9) | Sole dark-adapted vision impairment (%) (n = 5) | Dual hearing and vision impairment (%) (n = 2) | P-value |
| Sex | | | | | | |
| Male | 53.3 | 54.2 | 55.6 | 40.0 | 50.0 | 0.95 |
| Female | 46.7 | 45.8 | 44.4 | 60.0 | 50.0 | |
| Body weight | | | | | | |
| <50lb | 33.3 | 35.6 | 0 | 40.0 | 50.0 | **0.02** |
| ≥50lb | 66.7 | 64.4 | 100.0 | 60.0 | 50.0 | |
| Age | | | | | | |
| <8 years | 50.7 | 59.3 | 22.2 | 20.0 | 0.0 | **0.02** |
| ≥ 8 years | 49.3 | 40.6 | 77.8 | 80.0 | 100.0 | |

CADES [1] were included as additional covariables for the larger questionnaire assessment dataset (Tables 4 and 5).

Sex had no significant relationship to prevalence of sensory impairments using either clinical or questionnaire assessment. Body weight had an inconsistent relationship to prevalence of sensory impairments: it had no relationship to prevalence of clinical sensory impairments (light-adapted responses, Table 2) but was significantly related to prevalence of clinical sensory

**Table 4. Questionnaire evaluation of the prevalence of hearing, cognition and bright light visual behavior.** Questionnaire evaluated characteristics of dogs with no sensory impairments, sensory impairments (hearing, bright light visual behavior) and dual sensory impairments (n = 238). Cognitive impairment classified as CADES category >0 (mild, moderate or severe).

| | Total (%) | Sensory impairment | | | | |
| --- | --- | --- | --- | --- | --- | --- |
| | | No sensory impairment (%) (n = 174) | Sole hearing impairment (%) (n = 29) | Sole vision impairment (%) (n = 18) | Dual hearing and vision impairment (%) (n = 17) | P-value |
| Sex | | | | | | |
| Male | 48.3 | 48.3 | 51.7 | 38.9 | 52.9 | 0.82 |
| Female | 51.7 | 51.7 | 48.3 | 61.1 | 47.1 | |
| Body weight | | | | | | |
| <50lb | 52.5 | 47.7 | 58.6 | 61.1 | 82.4 | **0.03** |
| ≥50lb | 47.5 | 52.3 | 41.4 | 38.9 | 17.7 | |
| Age | | | | | | |
| <8 years | 50.8 | 59.2 | 44.8 | 22.2 | 5.9 | **< 0.001** |
| ≥ 8 years | 49.2 | 40.8 | 55.1 | 77.8 | 94.1 | |
| Lifestage | (n = 147) | (n = 108) | (n = 18) | (n = 17) | (n = 9) | |
| Young adult | 24.5 | 29.6 | 16.7 | 8.3 | 0.0 | |
| Mature adult | 24.5 | 28.7 | 16.7 | 16.7 | 0.0 | **< 0.001** |
| Senior | 29.3 | 29.6 | 16.7 | 58.3 | 11.1 | |
| Geriatric | 21.8 | 12.0 | 50.0 | 16.7 | 88.9 | |
| Cognitive impairment | | | | | | |
| No | 67.4 | 76.3 | 66.7 | 41.2 | 5.9 | **< 0.001** |
| Yes | 32.6 | 23.7 | 33.3 | 58.8 | 94.1 | |

**Table 5. Questionnaire evaluation of the prevalence of hearing, cognition and dim light visual behavior.** Questionnaire evaluated characteristics of dogs with no sensory impairments, sensory impairments (hearing, dim light visual behavior) and dual sensory impairments (n = 227). Cognitive impairment classified as CADES category >0 (mild, moderate or severe).

| | Total (%) | Sensory impairment | | | | |
|---|---|---|---|---|---|---|
| | | No sensory impairment (%) (n = 167) | Sole hearing impairment (%) (n = 29) | Sole vision impairment (%) (n = 17) | Dual hearing and vision impairment (%) (n = 14) | P-value |
| Sex | | | | | | |
| Male | 49.3 | 49.7 | 65.5 | 29.4 | 35.7 | 0.08 |
| Female | 50.7 | 50.3 | 34.5 | 70.6 | 64.3 | |
| Body weight | | | | | | |
| <50lb | 53.3 | 48.5 | 62.1 | 58.8 | 85.7 | **0.03** |
| ≥50lb | 46.7 | 51.5 | 37.9 | 41.2 | 14.3 | |
| Age | | | | | | |
| <8 years | 52.0 | 60.5 | 41.4 | 23.5 | 7.1 | **< 0.001** |
| ≥ 8 years | 48.0 | 39.5 | 58.6 | 76.5 | 92.9 | |
| Lifestage | (n = 141) | (n = 105) | (n = 18) | (n = 10) | (n = 8) | |
| Young adult | 24.8 | 29.5 | 16.7 | 10.0 | 0.0 | |
| Mature adult | 24.8 | 30.4 | 11.1 | 0.0 | 12.5 | **< 0.001** |
| Senior | 29.8 | 30.4 | 22.2 | 60.0 | 0.0 | |
| Geriatric | 20.6 | 9.5 | 50.0 | 30.0 | 87.5 | |
| Cognitive impairment | | | | | | |
| No | 67.3 | 74.7 | 55.2 | 52.9 | 21.4 | **< 0.001** |
| Yes | 32.8 | 25.3 | 44.9 | 41.2 | 78.6 | |

impairments (dark-adapted retinal responses, Table 3) and questionnaire assessed sensory impairments (Tables 4 and 5). Age had a consistent relationship to prevalence of sensory impairments as assessed both clinically and by questionnaire. By both methods of assessment, older dogs (≥ 8 years) had a higher prevalence of sensory impairments than younger dogs (<8 years). The overall prevalence of no sensory impairments was consistent between outcome measures–no sensory impairments were detected in 59–62% of younger dogs (<8 years) and 39–41% of older dogs (≥ 8 years). Similarly, using assigned lifestage in the questionnaire-assessed population (Tables 4 and 5), geriatric dogs had 10–12% prevalence of no sensory impairments, compared with 30% of young adult dogs. There were significant relationships between prevalence of questionnaire assessed sensory impairments and cognitive impairment, whereby cognitive impairment was more commonly described in dogs with sensory impairments and was most commonly reported in dogs with dual hearing/vision impairments (Tables 4 and 5).

## Relationships between sensory and cognitive functions

We performed analyses on the questionnaire-assessed group that determined the prevalence of cognitive impairment in dogs with different levels of sensory impairments. Data analyses were performed for sex, age and body weight (Table 6 bright light vision, Table 7 vision in near darkness). Overall, prevalence of cognitive impairment was higher in dogs with sensory impairments, and highest in dogs with dual hearing and vision impairments. Most consistently, older dogs and smaller dogs had significant variance in prevalence of cognitive impairment with varying sensory impairment status. Cognitive impairment was more

**Table 6. Questionnaire assessed prevalence of cognitive impairment in dogs with sensory impairments (hearing, bright light visual behavior) and dual sensory impairments (n = 230).** Cognitive impairment classified as CADES category >0 (mild, moderate, or severe).

|  | All dogs | All male | All female | <8 years | ≥ 8 years | <50lb | ≥ 50lb |
|---|---|---|---|---|---|---|---|
|  | (n = 230) | (n = 115) | (n = 123) | (n = 117) | (n = 113) | (n = 124) | (n = 106) |
| No impairment (n = 169) | 23.7 | 25.6 | 21.8 | 22.1 | 24.3 | 25.6 | 21.8 |
| Sole hearing impairment (n = 27) | 33.3 | 23.7 | 41.7 | 23.1 | 37.5 | 29.4 | 40.0 |
| Sole vision impairment (n = 17) | 58.8 | 66.7 | 54.6 | 75.0 | 50.0 | 54.5 | 66.7 |
| Dual hearing and vision impairment (n = 17) | 94.1 | 100.0 | 87.5 | 100.0 | 93.8 | 100.0 | 66.7 |
| P value for differences in cognitive impairment by sensory impairment | <0.001 | <0.001 | <0.001 | 0.03 | <0.001 | <0.001 | 0.02 |

prevalent in older dogs with both hearing and vision impairments, and in smaller dogs with both hearing and vision impairments.

Because older age was the most consistent and significantly associated covariable relating to cognitive impairment and sensory impairment relationships, we calculated odds ratios for risk of cognitive impairment in older (≥ 8 years) dogs with different degrees of sensory impairment, using dogs with no impairments as the reference population (Table 8). Sole hearing and vision impairments moderately increased risk of cognitive impairment in this population (1.3 odds ratio), but dual hearing and visual impairments substantially increased risk of cognitive impairment (1.8–2.0 odds ratio).

## Discussion

We have described relationships between dog age, and impairments in hearing and vision. We developed cutoffs for hearing and vision impairments (measured by both clinical methods and by proxy owner questionnaire), using data obtained from young dogs. Using these cutoffs, we determined that hearing and vision impairments were more prevalent in dogs aged 8 years or older, as measured by both clinical and proxy methods. Similarly, cognitive impairment (measured by a validated proxy dog owner questionnaire) was more prevalent in dogs aged 8 years or older. Combined sensory impairments of both hearing and vision were associated with an approximately 2-fold higher risk of cognitive impairment in these older dogs.

Age was associated with hearing impairment in dogs. We performed a detailed clinical electrophysiological assay of dog hearing, using brainstem auditory evoked potential (BAEP) measures in response to broadband click stimuli. We identified inconsistent associations between dog age and measurements of amplitude and latency of BAEP waveforms in response to standardized stimulus intensities (90, 70, 50 dB nHL), limiting their potential application in the longitudinal study of dog aging and hearing loss. Conversely, the threshold of BAEP response was significantly associated with dog age, with older dogs demonstrating a

**Table 7. Questionnaire assessed prevalence of cognitive impairment in dogs with sensory impairments (hearing, very dim light visual behavior) and dual sensory impairments (n = 227).** Cognitive impairment classified as CADES category >0 (mild, moderate, or severe).

|  | All dogs | All male | All female (n = 115) | <8 years | ≥ 8 years | <50lb | ≥ 50lb |
|---|---|---|---|---|---|---|---|
|  | (n = 227) | (n = 112) |  | (n = 118) | (n = 109) | (n = 121) | (n = 106) |
| No impairment (n = 167) | 25.3 | 26.5 | 24.1 | 24.8 | 25.8 | 28.4 | 22.3 |
| Sole hearing impairment (n = 29) | 44.8 | 47.4 | 40.0 | 33.3 | 52.9 | 44.4 | 45.5 |
| Sole vision impairment (n = 17) | 47.1 | 60.0 | 41.7 | 25.0 | 53.9 | 40.0 | 57.1 |
| Dual hearing and vision impairment (n = 14) | 78.6 | 80.0 | 77.8 | 0.0 | 84.6 | 83.3 | 50.0 |
| P value for differences in cognitive impairment by sensory impairment | <0.001 | 0.02 | 0.007 | 0.84 | <0.001 | 0.003 | 0.05 |

**Table 8. Odds ratio (confidence intervals) of any cognitive impairment (CADES > 0) for dogs aged > 8 years, stratified by bright/dark vision and owner-questionnaire.** * indicates significance as the 95% confidence interval does not include 1.

|  | Hearing vs. bright vision (n = 113) | Hearing vs. dark vision (n = 109) |
|---|---|---|
| No impairment | 1.00 (Ref) | 1.00 (Ref) |
| Hearing impairment only | 1.17 | 1.31 |
|  | (0.91, 1.49) | (1.03, 1.67)* |
| Vision impairment only | 1.34 | 1.32 |
|  | (1.04, 1.73)* | (1.01, 1.74)* |
| Both hearing and vision impairment | 2.00 | 1.80 |
|  | (1.58, 2.53)* | (1.37, 2.36)* |

consistently higher BAEP response threshold. BAEP threshold testing is used clinically in humans where behavioral audiometry testing cannot be performed [50], which also applies to our nonverbal dog population. The range of BAEP thresholds in our study were substantially lower than those previously reported in a study of geriatric dogs [15]. Reasons for this could include differences in study population age demographics, units of measurement of stimulus intensity, the range of stimulus intensities presented, or the concurrent administration of medications. Our described population spanned lifestages 2–5, whereas Fefer et al. [15] described results from predominantly dogs at lifestage 5; the median age for dogs in their study was 156 months, whereas our median age was 89 months (for clinically examined dogs) and 95 months (for questionnaire-assessed dogs). We presented stimuli in normalized Hearing Level (nHL) units in our study, and found a similar BAEP threshold range to a study performed in dogs with otitis externa that also presented stimuli in nHL units [38]. The study by Fefer et al. [15] presented stimuli measured in Sound Pressure Level (SPL) units, which we found on our instrumentation presents stimuli >20dB lower intensity than nHL calibrated stimuli. Our measured thresholds were estimated to within a 5dB nHL sensitivity range, versus a 20dB sPL range in Fefer et al. 2022 [15]. We did not use any anxiolytic or sedative medications in any dog in our study because of their documented effects on neurologic outcome measures in dogs [17, 36, 37]. The exact effects of these medications on BAEP have not been described. In addition to electrophysiological evidence of elevated hearing thresholds in older dogs, we also identified age-associated hearing loss using a previously validated proxy dog hearing function questionnaire [38]. This questionnaire had high specificity, albeit with lower sensitivity compared with the gold standard of BAEP threshold determination. The outcomes of our sensitivity and specificity analysis using this questionnaire were similar to those reported in the original publication of this questionnaire using dogs with acquired hearing loss due to otitis externa [38]. The reported study had an 83% sensitivity and 94% specificity for hearing loss >40dB nHL, whereas we found a 75% sensitivity and 90% specificity. Similarly in humans, self-report of hearing impairment is a specific and relatively sensitive way to identify patients with clinically meaningful hearing loss [51].

We also confirmed that dogs are susceptible to age-related vision impairment and developed cutoffs for both vision and hearing impairment. Similar to findings in previous publications [16, 17, 52, 53], our study found that dog age was associated with vision impairment, both when assessed by clinical measurement (electroretinography) and by proxy questionnaire (dogVLQ). Because our sample contained a representative sample from young, healthy dogs, we were able to use these data to calculate projected impairment cutoffs to estimate the prevalence of sensory (hearing and vision) impairments. Our criteria for determining a hearing impairment cutoff were determined objectively using lifestage 2 responses, and for the hearing

questionnaire were similar to the original study [38]; owners had to have selected responses to 2 questions that indicated their dog had hearing loss. The use of cutoffs for human hearing and vision impairments is common and has utility in both clinical practice and epidemiological studies [54–56]. Using our calculated impairment cutoffs, we identified relationships between the prevalence of sensory impairments and dog age. Using clinical and questionnaire outcome measures, dogs aged 8 years or older had a higher prevalence of hearing impairment (24% clinical, 27% questionnaire) compared with younger dogs (5% clinical, 12% questionnaire). As previously mentioned, our sample was relatively young, and the age of 8 years may not reflect the best age beyond which hearing impairment is most common. Our lifestage data support this concept, as geriatric dogs (beyond the end of expected lifespan) had a significantly higher BAEP response threshold than all other lifestages. For vision, using clinical and questionnaire measures, dogs aged 8 years or older had a higher prevalence of vision impairments (14–24% clinical, 24–26% questionnaire) compared with younger dogs (0–2% clinical, 4% questionnaire). This finding was supported by our lifestage analysis where both senior and geriatric dogs had lower ERG amplitudes than young adult dogs. Our work indicates a potential earlier onset of vision impairment compared with hearing impairment, although further longitudinal study would be necessary to confirm, as we only performed cross-sectional analysis. A combination of both hearing and vision impairments was commonly identified in older dogs using both clinical and questionnaire measures. Older dogs aged 8 years or older had substantially higher prevalence of dual sensory impairment (7–14% clinical, 14–16% questionnaire) compared with younger dogs (0% clinical, 0.8% questionnaire); approximately half of dogs in this age category with vision impairment also had hearing impairment, and vice versa. This statistic is highly similar to that found in humans, whereby an estimated 40% of people with low vision also have hearing impairment [57]. Dual sensory impairment in humans affects independence and social interaction and reduces overall quality of life [58, 59]. Blindness in dogs has perceived effects on emotions including perceived depression, increased owner dependency, and reliance on other senses, primarily hearing and olfaction [60]. The effects of dual sensory impairments on quality of life in older dogs has not been described, but in young dogs with both congenital hearing and vision impairments, behavioral abnormalities were commonly identified, and participation in dog sports was diminished [61], indicating the possibility of negative effects on quality of life in aging dogs that develop sensory impairments later in life. Our work highlights that older dogs with one sensory impairment are at high risk for other sensory impairments. While we only studied 2 senses in this work, reports in large scale human studies indicate that when all five senses are included (vision, hearing, smell, touch, taste), the prevalence of dual sensory impairment is over 60% in older adults [62].

We identified relationships between hearing and vision impairments and cognitive impairments in dogs. A higher prevalence of cognitive impairment was most associated with sensory impairments in dogs aged 8 years or older, and in smaller dogs weighing less than 50lb. In older dogs, there were small, independent associations found between sole vision and hearing impairments and risk for cognitive impairment, but a stronger risk association was present when dogs had dual sensory impairment. Previous studies have determined that worse cognitive function in dogs is associated with worse hearing [15], vision, and sense of smell [4]. Similarly, eye and ear disorders were independently associated with higher risk for cognitive impairment in a large population-based study of dogs in the United States [5], although exact disorders were not described, nor how impaired each sense was in association with these disorders. No studies to-date have examined the relationship between cognition and combined impairments of multiple senses in dogs. There is strong evidence of such a relationship in humans. Compared with no sensory impairment or sole sensory impairment, people with dual or multiple sensory impairments are predisposed to a greater degree of cognitive decline [28,

63, 64], and more extensive cortical amyloid beta deposition [65], a critical pathologic hallmark of Alzheimer's disease and related dementias. However, the relationship between sensory and cognitive impairments is not definitively unidirectional–for example, a bidirectional relationship between impairments in vision and cognition has been suggested by metanalysis of human studies [21]. Patients with dementia are at increased risk of developing vision [21] and hearing impairments [22]. Conversely, there is a higher risk of dementia in patients with sensory morbidities including vision impairment [23–28], hearing impairment [28–31], and olfactory impairment [32, 33]. Because the relationship between dementia and sensory impairment is bidirectional, this supports the "common cause" hypothesis which suggests that shared pathways of external factors influence the incidence of multiple disorders [66]. The study of these external factors may be most efficiently studied using a relevant animal model, such as dogs. Further longitudinal studies in dogs are needed to determine if sensory impairments predict worse cognitive decline in older dogs or vice versa, to begin to elucidate potential relationships.

We found a significant relationship between dog size and subjective (questionnaire-assessed) sensory impairment. These analyses suggested that smaller dogs had a greater prevalence of sensory impairments, and relationships between sensory and cognitive impairments were stronger in smaller dogs. However, our clinical outcome measures did not confirm the questionnaire-assessed sensory impairment findings. In fact, we identified overall higher prevalence of sensory impairments in larger dogs, although with the limitation that we under-sampled small dogs in our clinically assessed subgroup which limited statistical power. Other studies have also identified that smaller dogs have higher prevalence of cognitive impairment as assessed by questionnaire [3, 67], but no studies have validated this finding in clinical studies.

Our findings should be considered in context of some limitations. Firstly, self-report questionnaires regarding impairment in human epidemiological studies are considered less reliable than clinical outcome measures, and our questionnaires require proxy reporting as self-report is not feasible in dogs. However, all our utilized questionnaires have been compared to clinical outcome measures and while they are relatively insensitive to the extent of impairment we determined using clinical measures, they have suitable validity to detect meaningful impairment. Many dogs in the study had neurologic abnormalities identified on clinical examination, which may affect owners' interpretation of hearing, vision and cognitive impairment in the home environment. However, these abnormalities were overall mild and potentially transient. The prevalence of neurologic abnormalities in our group of dogs was in line with prevalence in a similar study in apparently healthy dogs [36]. Secondly, our cutoff values were based on data generated from young dogs, and in order to fully generalize to the dog population require further validation on a larger scale and with further supporting clinical validation. Thirdly, with respect to BAEP threshold determination, broadband click stimuli may not be the most optimal stimulus type with which to study age-associated hearing loss in dogs. In humans with cochlear hearing loss, broadband click BAEP is most associated with loss of low frequency hearing in the 2–4 kHz range, [68]. Studies in dogs have identified that age is associated with loss of higher frequency hearing, with the greatest loss noted in the 8-32kHz frequency range in dogs aged over 8–9 years [20]. It is therefore possible that our study lacked sensitivity to high frequency hearing loss, and future studies should employ frequency-specific stimuli to evaluate age-related hearing loss in dogs. Finally, based on our small sample of clinically evaluated dogs, we were unable to confirm our questionnaire-based findings using objective clinical evaluation of cognitive function, and future studies will be necessary to confirm our findings using objective clinical outcome measures.

## Supporting information

**S1 Fig. BAEP amplitude in association with age in dogs.** Amplitude of brainstem auditory evoked potential measurements for wave I and wave V in association with dog age. Simple linear regression lines are shown for illustration purposes only. Nonparametric Spearman rank r values and p values are shown, in association with dog age. Male dogs are illustrated as crosses, female dogs as circles.
(TIF)

**S2 Fig. BAEP latency in association with age in dogs.** Latency of brainstem auditory evoked potential measurements for wave I and wave V in association with dog age. Simple linear regression lines for significantly associated parameters are shown for illustration purposes only. Male dogs are illustrated as crosses, female dogs as circles. Nonparametric Spearman rank r values and p values are shown, in association with dog age. Male dogs are illustrated as crosses, female dogs as circles.
(TIF)

**S3 Fig. Relationship between nHL and SPL.** Correlation between assigned BAEP nHL and measured SPL. R2 >0.99, P<0.0001, simple linear regression. X axis intercept -23.73, Y axis intercept 24.14, equation Y = 1.017*X + 24.14.
(TIF)

**S4 Fig. Visual outcome measures in association with age in dogs.** Vision outcome measures in relation to dog age in the study population for electroretinography (upper graphs) and questionnaire (lower graphs) outcome measures. Spearman rank correlation statistics are shown for continuous variables, and a regression line is shown for illustration purposes. For categorical data, Kruskal Wallis statistical significance is shown. * P<0.05, *** P<0.001. Calculated cutoffs are shown using a dotted horizontal line. Male dogs are illustrated as crosses, female dogs as circles.
(TIF)

**S1 Raw data. Spreadsheet containing raw data used for statistical analysis.**
(XLSX)

## Acknowledgments

The authors thank Dr. Helena Rylander and Dr. Natalia Zidan for completing neurologic examinations on clinical participants, Aaron Kopydlowski, Amelia Corona and Amy Elbe for assistance with recruitment and dog restraint during clinical visits, Hannah Lillesand for assisting with ERG analysis. We thank G. Nike Gnanateja for performing measurements and calculations for the nHL/SPL conversion.

## Author Contributions

**Conceptualization:** Starr Cameron, Freya M. Mowat.

**Data curation:** Ryan G. Hopper, Rachel B. Bromberg, Michele M. Salzman, Freya M. Mowat.

**Formal analysis:** Ryan G. Hopper, Rachel B. Bromberg, Kyle D. Peterson, Freya M. Mowat.

**Funding acquisition:** Starr Cameron, Freya M. Mowat.

**Investigation:** Ryan G. Hopper, Rachel B. Bromberg, Michele M. Salzman, Callie Rogers, Starr Cameron, Freya M. Mowat.

**Methodology:** Michele M. Salzman, Kyle D. Peterson, Starr Cameron, Freya M. Mowat.

**Project administration:** Starr Cameron, Freya M. Mowat.

**Resources:** Kyle D. Peterson, Freya M. Mowat.

**Supervision:** Starr Cameron, Freya M. Mowat.

**Visualization:** Freya M. Mowat.

**Writing – original draft:** Ryan G. Hopper, Rachel B. Bromberg, Kyle D. Peterson.

**Writing – review & editing:** Michele M. Salzman, Kyle D. Peterson, Callie Rogers, Starr Cameron, Freya M. Mowat.

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
