## [Decision Letter · Decision Letter 0]

30 Jul 2024

PONE-D-24-17815Dual sensory impairments in companion dogs: prevalence and relationship to cognitive dysfunction.PLOS ONE

Dear Dr. Mowat,

Thank you for submitting your manuscript to PLOS ONE. After careful consideration, we feel that it has merit but does not fully meet PLOS ONE’s publication criteria as it currently stands. Therefore, we invite you to submit a revised version of the manuscript that addresses the points raised during the review process. I have carefully read the manuscript which I find very interesting. I agree with the comments of the two reviewers that I think would improve the manuscript for readers. If you can attend to their points (expecially the problem with the validation of the questionnaires) I am confident I will be able to accept the manuscript.

We look forward to receiving your revised manuscript.

Kind regards,

Maria Santacà

Academic Editor

PLOS ONE

Journal Requirements:

"This work was funded by the National Institutes of Health (K08EY028628, R01AG082907, P30EY016665), the Morris Animal Foundation Mark L. Morris Jr. Investigator Award (D23CA-510), donor funds to the UW-Madison Department of Ophthalmology and Visual Sciences, an unrestricted grant from Research to Prevent Blindness, Inc. to the UW-Madison Department of Ophthalmology and Visual Sciences, and the UW-Madison Women in Science and Engineering and Leadership Institute Vilas Life Cycle Professorship. "

Reviewers' comments:

Reviewer's Responses to Questions

**Comments to the Author**

1. Is the manuscript technically sound, and do the data support the conclusions?

Reviewer #1: Yes

Reviewer #2: Yes

2. Has the statistical analysis been performed appropriately and rigorously? 

Reviewer #1: Yes

Reviewer #2: Yes

3. Have the authors made all data underlying the findings in their manuscript fully available?

Reviewer #1: Yes

Reviewer #2: Yes

4. Is the manuscript presented in an intelligible fashion and written in standard English?

Reviewer #1: Yes

Reviewer #2: Yes

5. Review Comments to the Author

Reviewer #1: The study assesses sensory and cognitive deficits in dogs using both physiological measurements (brainstem auditory evoked potential - BAEP and electroretinography - ERG) and questionnaire assessments. The authors implied at the very beginning that they would explain (make it understood) the relationship between sensory and cognitive function (l20-22: "Dog owners often describe impairments in multiple sensory functions, yet the relationships between sensory and cognitive function in older dogs are not well understood.") However, this explanation did not occur. What the reader received were correlations between impairments that, I think, are already well-known, even among dog owners, as mentioned in the abstract.

My main expectation was that the extensive objective clinical data (which is truly great and important) would validate the subjective assessments of sensory deficits and cognitive capabilities. A questionnaire that reliably measures impairments would be very useful for veterinarians, researchers, and owners alike. However, this question was not the focus of the present analyses. The highlighted finding is the significant association between dual sensory impairments (hearing and vision) and increased prevalence of cognitive impairments in older dogs, but these findings, are neither novel nor particularly surprising and there is no need for physiological measurements to achieve these results.

With some difficulties, I could find the correlations between subjective and objective measures, but they were rather weak, as in the case of hearing (Spearman r = -0.27, p = 0.02), or nonexistent, as in the case of vision (at least, "data is not shown" - l292). In the latter case, there was a discrepancy between the subjective and objective measures of sensory impairments, particularly in smaller dogs, where questionnaire results indicated a higher prevalence of sensory impairments compared to clinical evaluations. Unfortunately, this raises questions about the reliability and validity of the applied questionnaires. Although it was mentioned in the text that these questionnaires are validated, perhaps this validation process was not adequate. This issue (i.e. what is the problem with the questionnaires and how could they be developed to improve validity) should be discussed in more detail in the manuscript.

I suggest adding paragraphs in the Methods and Results sections that explicitly investigate the relationship between the questionnaire data and the physiological measures.

The methodological section lacks adequate detail, particularly with regard to the cognitive dysfunction and sensory impairments questionnaires. Only references are mentioned.

Discussion: As promised in the abstract, hypotheses on the biological background ("understanding") of the possible reciprocal relationship between sensory and cognitive deficits would improve the text.

Other comments:

- l20 (Canis...) - The Latin name should begin with a capital C.

- l49 and elsewhere: "cognitive impairment" is used as a synonym for "cognitive dysfunction" in the text. I suggest consistently using the same terminology throughout the manuscript to ensure clarity.

- l84 Study participants: Add the number of participants in this section.

- l98-100: Provide more information about the questionnaires (e.g., number of questions, example questions, how these tools were validated), so readers do not need to look them up in the references. This will also help clarify whether the present study could improve the validation, especially if there were gaps in previous literature. If gaps are identified, mention them in the Introduction.

- l150: Include the weight category of large and small dogs, so readers do not need to look it up in the reference.

- l160: "largest group of young dogs" - Clarify that "large" refers to the sample size and not the body size, if this is the case.

- l169-170: "using standard methods" - Briefly describe what these methods are.

- l185: Explain what can be learned from the list of diseases. How representative are these dogs? What kind of population do they represent?

- l483: Clarify if there was financial support for this study.

Reviewer #2: In this manuscript the authors assessed dog vision and hearing clinically and via questionnaires, to assess the relationships between vision and/or hearing impairments and cognitive impairments. Their results indicate a potential earlier onset of vision impairment compared with hearing impairment.

The authors also established sensory impairment cutoffs based on 161 young dogs then used this to evaluate sensitivity and specificity of the questionnaire. Their goal is to provide a methodology to detect impairment using objective clinical tests using standard methods.

The research question is timely and the approach of the study is also valid. I applaud the authors for administering any sedation or anxiolytics to get better quality data and that they assigned the dogs by life stages.

The presentation of the results is very informative and detailed.

I only have a few minor comments:

line 107 A subset of dogs (n = 91) participated. --It is not clear here in the text what does this refer to, subset of what?

line 146 dak adaptation --typo

The sample size is too small to evaluate the effect of life stages, body size or cephalic index (and their possible interactions with sufficient resolution), but the study is excellent for validating the questionnaires.

What is the difference between table 2 and 3? They seem to be same data with minor differences ( one with n=75 the other with n=76)

I have the same question for table 4 and 5. It is not clear what is the difference between them based on the legend,, apart from sample size (238 vs 227)

6. PLOS authors have the option to publish the peer review history of their article (what does this mean?). If published, this will include your full peer review and any attached files.

Reviewer #1: No

Reviewer #2: **Yes: **Dora Szabo, PhD

---

## [Author Response · Author response to Decision Letter 0]

13 Aug 2024

A comprehensive response to reviewer document has been uploaded and submitted

---

## [Decision Letter · Decision Letter 1]

29 Aug 2024

Dual sensory impairments in companion dogs: prevalence and relationship to cognitive impairment.

PONE-D-24-17815R1

Dear Dr. Mowat,

We’re pleased to inform you that your manuscript has been judged scientifically suitable for publication and will be formally accepted for publication once it meets all outstanding technical requirements.

Kind regards,

Maria Santacà

Academic Editor

PLOS ONE

Additional Editor Comments (optional):

Reviewers' comments:

Reviewer's Responses to Questions

**Comments to the Author**

1. If the authors have adequately addressed your comments raised in a previous round of review and you feel that this manuscript is now acceptable for publication, you may indicate that here to bypass the “Comments to the Author” section, enter your conflict of interest statement in the “Confidential to Editor” section, and submit your "Accept" recommendation.

Reviewer #1: All comments have been addressed

Reviewer #2: All comments have been addressed

2. Is the manuscript technically sound, and do the data support the conclusions?

Reviewer #1: Yes

Reviewer #2: Yes

3. Has the statistical analysis been performed appropriately and rigorously? 

Reviewer #1: Yes

Reviewer #2: Yes

4. Have the authors made all data underlying the findings in their manuscript fully available?

Reviewer #1: Yes

Reviewer #2: Yes

5. Is the manuscript presented in an intelligible fashion and written in standard English?

Reviewer #1: Yes

Reviewer #2: Yes

6. Review Comments to the Author

Reviewer #1: (No Response)

Reviewer #2: I would like to than the authors you for addressing my comments via their revised manuscript. The tables are now much easier to interpret.

7. PLOS authors have the option to publish the peer review history of their article (what does this mean?). If published, this will include your full peer review and any attached files.

Reviewer #1: No

Reviewer #2: **Yes: **Dora Szabo

---

## [Editor Report · Acceptance letter]

6 Sep 2024

PONE-D-24-17815R1 

PLOS ONE

Dear Dr. Mowat, 

I'm pleased to inform you that your manuscript has been deemed suitable for publication in PLOS ONE. Congratulations! Your manuscript is now being handed over to our production team.

Kind regards, 

on behalf of

Dr. Maria Santacà 

Academic Editor

PLOS ONE